# Modulation of Visual Responses and Ocular Dominance by Contralateral Inhibitory Activation in the Mouse Visual Cortex

**DOI:** 10.3390/ijms24065750

**Published:** 2023-03-17

**Authors:** Wei Wu, Lei Li, Yueqin Liu, Luwei Kang, Hui Guo, Chenchen Ma, Yupeng Yang

**Affiliations:** Hefei National Laboratory for Physical Sciences at the Microscale, Chinese Academy of Sciences Key Laboratory of Brain Function and Diseases, Division of Life Sciences and Medicine, University of Science and Technology of China, Hefei 230027, China

**Keywords:** intrinsic signal optical imaging, optogenetics, visual response, ocular dominance, interneuron, callosal projection

## Abstract

Both hemispheres connect with each other by excitatory callosal projections, and whether inhibitory interneurons, usually believed to have local innervation, engage in transcallosal activity modulation is unknown. Here, we used optogenetics in combination with cell-type-specific channelrhodopsin-2 expression to activate different inhibitory neuron subpopulations in the visual cortex and recorded the response of the entire visual cortex using intrinsic signal optical imaging. We found that optogenetic stimulation of inhibitory neurons reduced spontaneous activity (increase in the reflection of illumination) in the binocular area of the contralateral hemisphere, although these stimulations had different local effects ipsilaterally. The activation of contralateral interneurons differentially affected both eye responses to visual stimuli and, thus, changed ocular dominance. Optogenetic silencing of excitatory neurons affects the ipsilateral eye response and ocular dominance in the contralateral cortex to a lesser extent. Our results revealed a transcallosal effect of interneuron activation in the mouse visual cortex.

## 1. Introduction

Long-range inputs are thought to modulate neuronal responses by supplying contextual information. Running can double the visual response of visual cortex neurons by basal forebrain cholinergic innervation, while serotonergic input from the dorsal raphe nucleus suppresses spontaneous and evoked activity in the visual cortex [1,2]. Specifically, transcallosal projections link the two cerebral hemispheres to coordinate neural activity. Blockade of interconnection results in abnormal development and impairs interhemispheric transfer of sensory information in rodents [3,4,5]. In the visual cortex, silencing one hemisphere reduces the responsiveness of ipsilateral eye and shifts ocular dominance to the contralateral eye. Muscimol injection or cooling is usually used to silence the activity of all neurons in a manipulated cortex. However, complex local circuits are composed of excitatory and inhibitory cells [6]. The roles of specific types of neurons, especially interneurons, in callosal processing have not been defined.

Most transcallosal projection neurons are excitatory neurons in adulthood, although some callosal projection neurons are transiently gamma-aminobutyric acid (GABA)-containing neurons at birth [7,8]. In the visual cortex, callosal projection neurons show higher ipsilateral eye preference and target neurons with similar orientation selective properties. These callosal projections contribute to the response of the ipsilateral eye and are associated with ocular dominance changes. In rodents and humans, interhemispheric inhibition was observed extensively [9,10,11,12], which is in conflict with the lack of transcallosal inhibitory interneurons. These excitatory callosal projection cells target excitatory and inhibitory neurons in the contralateral cortex. Although there is no direct inhibitory innervation to the contralateral cortex, interhemispheric inhibition is probably mediated by excitatory projection neurons and their inhibitory targets [13,14]. However, whether local inhibitory interneurons impact the contralateral hemisphere through their innervation to excitatory projection neurons remains to be elucidated.

Ocular dominance refers to the binocular area preferring visual inputs from one eye to the other. Abnormal ocular dominance results in diseases such as amblyopia. Callosal input is involved in ocular dominance plasticity and contributes to normal ocular dominance. Blockade of callosal input shifts ocular dominance to the contralateral eye. GABAergic inhibition in the human visual cortex relates to eye dominance [15]. Preventing the elimination of chandelier cells, one kind of inhibitory neurons, results in deficient binocularity [16]. The effect of contralateral inhibitory neurons on ocular dominance is not well known.

To evaluate the effect of activation of different inhibitory neurons in one hemisphere on the visual response and ocular dominance of another hemisphere, we optogenetically activated distinctive interneurons in one hemisphere and recorded the response of the visual cortex in another hemisphere using intrinsic signal optical imaging. We found that optogenetic manipulation reduced the visually evoked response and shifted ocular dominance to the contralateral eye. Our findings revealed a transcallosal effect of activation of inhibitory neurons.

## 2. Results

### 2.1. Distinct Response Features to Activation of Specific Interneurons in the Cortex

Compared with excitatory neurons in the cortex, inhibitory neurons have highly heterogeneous cell types that are characterized by their unique protein expression and roles in neuronal circuits [17,18,19]. First, we optogenetically activated different inhibitory neurons in the visual cortex and recorded the response using intrinsic signal optical imaging (Figure 1a,b). *VGAT-ChR2-EYFP* mice expressing channelrhodopsin-2 (ChR2) directed to GABAergic interneuron populations by the mouse vesicular GABA transporter (VGAT) promoter were used for activation of global inhibitory cells [20]. Wild-type littermate mice were used as controls. To achieve selective activation of subpopulations of inhibitory cells, especially cells expressing pavalbumin (PV, Pvalb) or somatostatin (SST), *Pvalb-IRES-Cre* and *Sst-IRES-Cre* transgenic mice were crossed with line Ai32 (*Rosa*-CAG-LSL-ChR2(H134R)-EYFP-WPRE) [21,22]. The bred mice used in our study included C57BL/6J (control), *VGAT-ChR2_EYFP* (VGAT-ChR2), PV::Ai32 (PV-ChR2) and SST::Ai32 (SST-ChR2) mice.

In control mice, light cannot induce any reflection changes of illuminations except the fiber tip due to the contamination of light (Figure 1c). In VGAT-ChR2 mice, light induced a biphasic response pattern that contained an “activated region” (decrease in reflectance) and an “inhibited region” (increase in reflectance). The cortex near the fiber tip showed a positive signal after activation, while an inhibited region was observed in the cortex far from the fiber tip (Figure 1d). The activated region may be mediated by robust activation of inhibitory neurons and the inhibited region may be caused by a decrease in spontaneous activity. As the power of light increased, the cortex area showing a positive signal became larger (Figure 1d,g) and had an enhanced response magnitude (Figure 1d,h). These results indicate that the activation of inhibitory cells induced distinct response patterns and that the response magnitude depended on the intensity of the stimulus. However, different inhibitory interneuron types may show distinct sensitivity to light and have specific roles in the propagation of intrinsic signals. Next, we recorded the response induced by activation of the subpopulations of inhibitory neurons. In PV-ChR2 mice, the cortex near and far from the fiber tip both showed inhibited intrinsic signals induced by light stimulation (Figure 1d). However, the response amplitude was not significantly different between the two areas (Figure 1i,j). The cortex near the fiber tip showed an inhibited response, possibly because the parvalbumin-positive interneurons target all cell types containing themselves [6]. In SST-ChR2 mice, the response pattern was similar to that of VGAT-ChR2 mice, but the corresponding response amplitude was weaker (Figure 1e,i,j). Together these data show that activation of inhibitory neurons exhibits different response features.

### 2.2. Reduced Spontaneous Activity in the Binocular Area Contralateral to Stimulation

Next, we tested the effect of activation of inhibitory neurons in one hemisphere on another hemisphere. We arranged the optical fiber to target the right binocular area and recorded the light-induced response from left of the cortex (Figure 2a). In wild-type mice, light cannot induce any reflectance changes in the contralateral visual cortex (Figure 2b top). In VGAT-ChR2 mice, the activation of interneurons induced increased reflectance of illumination on the contralateral side. The binocular area in the contralateral cortex showed the maximal response (Figure 2b bottom). This indicated that the effect was carried by the corpus callosum because the binocular area and the border between the binocular area and second visual cortex are heavily innervated by callosal inputs [23]. When we located the tip of the optical fiber at different sites within the binocular area, the region of maximal response changed appropriately. In the binocular area, higher light power in the contralateral hemisphere induced a larger response magnitude (Figure 2d–f). Among the three transgenic mice, SST-ChR2 mice showed the lowest response variability to a particular power of light. 

### 2.3. The Impact of Contralateral Inhibitory Activation on Visually Evoked Response

To test whether activation of inhibitory neurons depressed the visually induced response in the contralateral visual cortex, we simultaneously activated the inhibitory neurons in the right visual cortex and recorded the visually evoked response in the left visual cortex (Figure 3a). In littermate controls, light stimulation in the contralateral cortex had no impact on the magnitude of the visually evoked response (Figure 3b,c). In VGAT-ChR2 mice, the response from the ipsilateral eye was abolished by contralateral light stimuli. The binocular area response evoked by the contralateral eye was decreased by light stimuli. The monocular response was changed to a lesser extent by light stimuli (Figure 3d). In PV-ChR2 mice, the impact of light stimulation on the contralateral hemisphere on the measurement of visual response was similar to that in VGAT-ChR2 mice, but the response variability was higher (Figure 3e). In SST-ChR2 mice, the impact of light stimuli on binocular response was similar to that in VGAT-ChR2 mice, but the monocular response was heavily reduced by light stimuli (Figure 3f). These results indicated that different inhibitory interneurons have similar impacts on the measurement of visual response.

Next, to consolidate the results from transgene mice, we injected a virus vector expressing chanlrhodopsin-2 after exposure to Cre recombinase into the binocular area of PV-Cre mice (Figure 4a left). To test whether there are callosally projecting PV neurons, we used two strategies: (1) the variant rAAV2/R, which permits robust retrograde access to projection neurons, was injected into the left visual cortex; (2) the variant rAAV2/9 was injected into the right visual cortex. PV mice injected with a virus vector expressing GFP were used as controls. Before imaging recording, we observed the expression of ChR2 (Figure 4a right). There were no labeled fibers in the visual cortex contralateral to the injection site. This indicated that PV interneurons cannot directly innervate the contralateral visual cortex. The imaging recording of both strategies is illustrated in Figure 4b. The data were pooled for analysis. In the control, contralateral light stimuli did not affect the visual response (Figure 4c). In mice expressing ChR2 in PV interneurons, the ipsilateral eye response and binocular response from the contralateral eye were decreased by light stimuli, and the monocular area response did not show an effect of light stimuli (Figure 4d). The results confirmed the phenomenon in transgene mice that contralateral inhibitory neuron activation modulated the visual response. 

### 2.4. Contralateral Inhibitory Activation Shifts Ocular Dominance to the Contralateral Eye

Having confirmed that inhibitory activation modulated the visual response of the contralateral hemisphere, we tested whether the activation of contralateral inhibitory interneurons can change ocular dominance. In control mice, the ocular dominance bias to the contralateral eye and this ocular dominance were not changed by light stimulation (Figure 5a). In transgene mice, the ocular dominance distribution was dominated by the contralateral eye and was shifted to the contralateral eye after contralateral inhibitory stimulation (Figure 5a), because the ipsilateral eye response was reduced more by contralateral inhibitory activation. Similar results were observed in virus-injected PV-Cre mice (Figure 5b).

### 2.5. The Effect of Inhibitory Neuron Activation Was Partially Mediated by Silencing of Excitatory Cells

To test whether the callosal inhibitory effect was mediated by excitatory cells, we optogenetically silenced α-Ca^2+^/calmodulin-dependent kinase II (CaMKIIα)-expressing neurons during visual stimuli. We injected a virus vector expressing halorhodopsin after exposure to Cre recombinase into the right binocular area of CaMKIIα-Cre mice. The fibers were observed in the contralateral visual cortex and ipsilateral superior colliculus (Figure 6a). We continuously silenced the right binocular area during visual stimuli and recorded the visual response to ipsilateral or contralateral eye stimuli in the left visual cortex (Figure 6b). Inactivation of contralateral excitatory neurons reduced the response to ipsilateral stimuli, but left the response of the contralateral eye unaltered. Next, we calculated the ODI and found that ODI increased after silencing of contralateral excitatory cells (Figure 6c,d). Compared with the activation of contralateral inhibitory cells, the silencing of excitatory cells affects the response and ocular dominance less. These results show that the modulation of the contralateral hemisphere by inhibitory activation is more than inhibition of local excitatory cells, which are callosally projecting.

## 3. Discussion

Whether activation of inhibitory neurons has an impact on the contralateral side was not determined. In this study, we used intrinsic signal optical imaging to evaluate the effect of activation of distinctive inhibitory neurons in one hemisphere on spontaneous and visually evoked responses of the contralateral cortex. First, we recorded the local responses induced by optogenetic activation of different inhibitory neurons and found that the response patterns showed different characteristics. However, all of these activations showed inhibitory effects on spontaneous activity in the contralateral visual cortex. Then, the impact on the visual response was characterized. The binocular response was affected mostly by contralateral inhibitory cell activation. This shifted ocular dominance to the contralateral eye. We confirmed this transcallosal effect on the visual response and ocular dominance in mice locally expressing ChR2 in PV neurons in the binocular area. Our results showed that activation of inhibitory neurons in one hemisphere has an impact on the contralateral side.

We recorded the response induced by light activation using intrinsic signal optical imaging. The fiber tip was not inserted into the visual cortex, but was positioned slightly above the thinned skull of the visual cortex. This manipulation will activate the visual cortex massively, but the depth affected is uncertain. Considering that some deep-layer GABAergic interneurons have dendritic arborization in the superficial cortex [6], light activation can be deeper than the penetration of light [24].

The activation patterns show center-periphery differences. In VGAT-ChR2 and SST-ChR2 mice, the center exhibited a positive intrinsic signal and the periphery showed a negative intrinsic signal after light stimulation. The peripheral negative intrinsic signal is probably due to an inhibitory effect, while the central positive signal can be induced by strong activation of inhibitory neurons. Another alternative explanation is that some inhibitory neurons innervate blood vessels [25,26,27,28,29]. In hippocampal microvessels, activation of the GABAA receptor elicits vasodilation [25]. In the cortex, optogenetic activation of inhibitory interneurons increases local blood flow through laser speckle imaging [29]. The origin of both the positive intrinsic signal near the fiber tip and the negative intrinsic signal around the fiber tip needs further investigation. In PV-ChR2 mice, the central and peripheral areas both show weak negative intrinsic signals, suggesting the inhibitory effect of PV interneurons on other cell types [6]. 

In these three types of transgene mice, activation of inhibitory neurons of the visual cortex in one hemisphere elicited a negative intrinsic signal on the other side. The response amplitude depends on the intensity of light. Although the response pattern evoked by light in the ipsilateral cortex displays different patterns, the negative intrinsic signals in the contralateral side are similar. This suggests that the effect was driven by inhibition on the contralateral side. The maximal response of the negative intrinsic signal was located in the binocular area, which is strongly innervated by contralateral inputs [30]. This suggests that the negative intrinsic signal was mediated by callosal projection.

We test the impact of negative intrinsic signals on the measurement of visually evoked responses. We found that the effect is variable according to the eye and receptive field of visual stimuli. The response of the ipsilateral eye was fully blocked by light stimulation of inhibitory neurons in the contralateral hemisphere. According to previous reports, callosal inputs contribute approximately one-third of the ipsilateral eye response [31]. This is consistent with the results of optogenetic silencing of excitatory neurons. The remaining portion may be mediated by other mechanisms. The binocular area response of the contralateral eye was partially blocked by the negative intrinsic signal. The response of the monocular area was affected less by the negative intrinsic signal. 

These results suggested that activation of inhibitory interneurons not only affected the cortex locally but also innervated the contralateral side transcallosally. The callosal effect has a different impact on the measurement of visually evoked responses.

## 4. Materials and Methods

### 4.1. Animals

Adult C57/BL6 male mice aged 8 weeks were purchased from the Model Animal Research Center of Nanjing University. *VGAT-ChR2-EYFP* (Jax No. 014548), Ai32 (Jax No. 024109) and *CaMKIIα-Cre* (Jax No. 005359) mice were a gift from Zhang Zhi lab. *PV-IRES-Cre* mice (Jax No. 008069), *SST-IRES-Cre* mice (Jax No. 013044) and Ai9 mice (Jax No.007914) were a gift from the Zhang Xiaohui lab. Mice were housed in ventilated cages under a 12 h dark-light cycle with food and water available. All experimental procedures were approved by the Institutional Animal Care and Use Committee at the University of Science and Technology of China.

### 4.2. Virus and Stereotaxic Surgery

The Cre-dependent adeno-associated viruses rAAV-EF1α-DIO-hChR2-EYFP-WPRE-hGH ployA, rAAV-EF1α-DIO-EYFP-WPRE-hGH ployA, rAAVretro-EF1α-DIO-hChR2-EYFP-WPRE-hGH ployA, rAAVretro-EF1α-DIO-EYFP-WPRE-hGH ployA and rAAV-EF1α-DIO-eNpHR3.0-EYFP-WPRE-hGH ployA were obtained from BrainVTA (BrainVTA, Wuhan, China). Mice were initially anesthetized with inhaled 4% isoflurane (RWD Life Science, Shenzhen, China) in oxygen, and maintained with 2% isoflurane. After fixation in a stereotaxic apparatus (RWD Life Science, Shenzhen, China), a small hole was drilled at the center of the binocular area determined by intrinsic signal optical imaging. A homemade microsyringe was connected to the UMP3 pump (World Precision Instruments inc., Sarasota, FL, USA) and a volume of 80–100 nL of virus was injected 0.5 mm below the pia at a rate of 30 nL/min. The pipette remained for 5 min after the completion of the injection. The scalp was sutured and the animal was returned to the cage until full consciousness.

### 4.3. In Vivo Photostimulation

The optical fiber tip was located at the center of the binocular area determined by intrinsic signal optical imaging. The optical fiber was coupled to a 473 nm or 593.5 nm solid-state laser (Changchun New Industries Optoelectronics Tech. Co., Ltd., Changchun, China). To avoid the direct effect of the laser on the contralateral hemisphere, a sheet made of cement (Zhangjiang Biological Materials Co. LTD, Shanghai, China) with carbon powder was placed on the midline. Blue light with variable intensity at a frequency of 20 Hz (duty ratio, 50%; duration, 4 s) was delivered. The power of blue light was adjusted by knob (0–20 mW). The intensity of the 20 mw indicator was 3.7 mW/mm^2^. The yellow light was delivered continuously at an intensity of 5 mW/mm^2^.

### 4.4. Intrinsic Signal Optical Imaging

After initial anesthesia with inhalation of 4% isoflurane in oxygen, the mice were fixed to a stereotaxic frame and maintained with 1.5% isoflurane. Body temperature was maintained at 37.5 °C through a heating pad. The skull above the visual cortex was thinned by a dental drill. To provide clear transparency during the recording, a drop of mineral oil was applied to the surface of the thinned skull. The vasculature was captured through 550 nm illumination. A Dalsa 1M60 CCD camera (Teledyne DALSA, Waterloo, ON, Canada) was used to collect the reflection of 720 nm illumination. The frame rate was 16 Hz and the size of the ROI was 160 × 160 pixels. The visual stimuli, laser and recording were triggered through a PCI card (National Instruments Corp., Austin, TX, USA). We used sinusoidal grating (spatial frequency: 0.05 cycles per degree, temporal frequency: 2 Hz, contrast: 1) as visual stimuli. The visual stimuli were represented in the binocular area (elevation: −5° to 15°; azimuth: −15° to 45°) and monocular area (elevation: 15° to 45°; azimuth: −15° to 45°). Before visual or light stimulation, the signal was recorded for 1 s as a blank, and the duration of visual stimuli or laser was 4 s with a 10 s recovery. The reflectance was calculated by (F − F0)/F0. We regard the 4–6 s reflection as response.

### 4.5. Histological Analyses for Virus Expression

The mouse was perfused transcardially with 0.1 M phosphate buffered saline (PBS) (Sangon Biotech, Shanghai, China) followed by 4% paraformaldehyde (Sangon Biotech, Shanghai, China). The brain was removed, postfixed overnight, cryoprotected with 30% sucrose and sectioned at 40 µm on a CM1950 microtome (Leica, Wetzlar, Germany). For detection of GFP expression, sections stained with 4,6-diamidino-2-phenylindole (DAPI) (Sangon Biotech, Shanghai, China) were mounted, coverslipped and visualized using an LSM 710 confocal laser scanning microscope (Carl Zeiss, Oberkochen, Germany).

### 4.6. Statistical Analyses

The statistical significance test was performed in GraphPad Prism (GraphPad Software, San Diego, CA, USA). Two-way repeated measures ANOVA and the two-tailed paired *t*-test were used. The summary of statistics is illustrated in Appendix A.

## 5. Conclusions

In this study, the effects on visual responses and ocular dominance of activation of contralateral inhibitory interneurons in the visual cortex were evaluated by intrinsic signal optical imaging. Optogenetic activation shows that different inhibitory neurons have distinctive response patterns. They all show inhibition of spontaneous activity in the contralateral visual cortex, and inhibition depends on the density of stimulations. Visually evoked responses were depressed by the activation of contralateral inhibitory neurons, and activation had the largest impact on the response from ipsilateral eyes. Thus, it shifts the ocular dominance to the contralateral eyes. Inhibition of excitatory neurons in the contralateral visual cortex reduced the ipsilateral eye response and moderately shifted the ocular dominance to the contralateral eye, suggesting that inhibition of excitatory neurons only partially mediates the effect of activation of inhibitory neurons. We conclude that activation of inhibitory neurons can modulate the visual response and ocular dominance in the contralateral visual cortex.

## Figures and Tables

**Figure 1 ijms-24-05750-f001:**
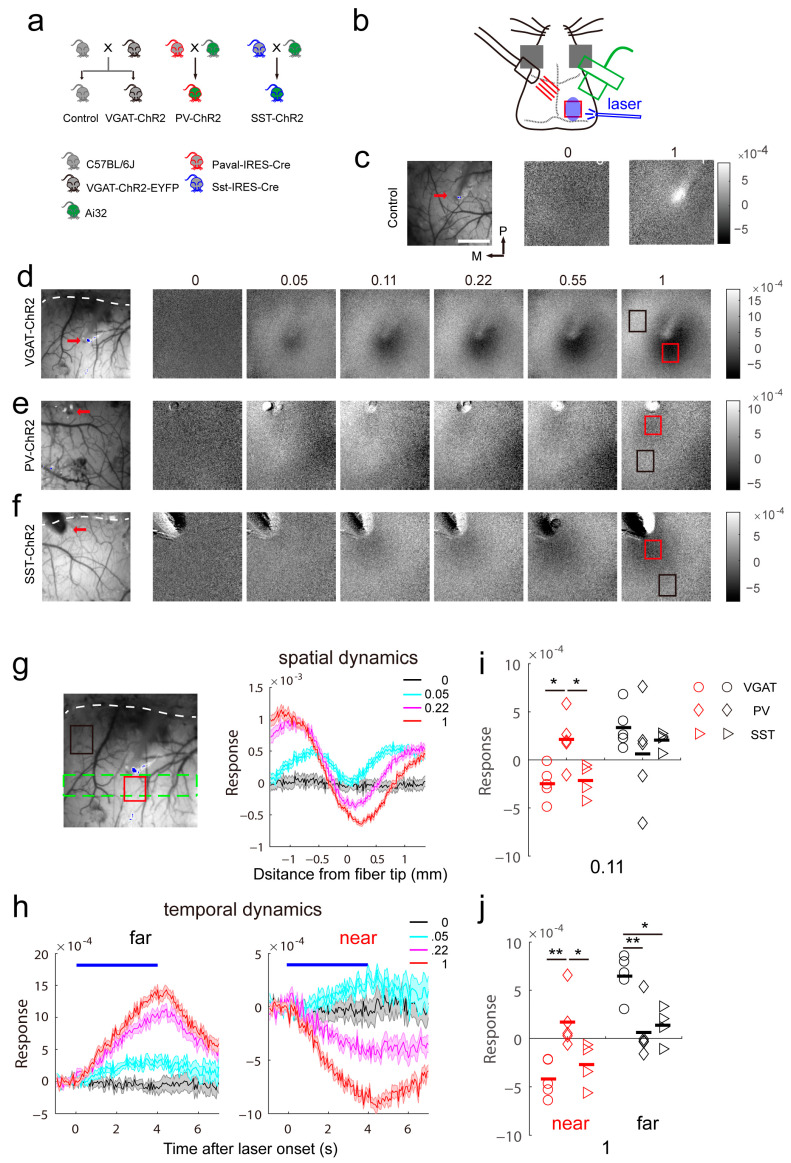
Differential local response induced by optogenetic stimulation of inhibitory neurons. (**a**) Mouse breeding strategy. (**b**) Schematic of intrinsic signal optical imaging and laser stimuli. Intrinsic signal optical imaging recorded the response to ipsilateral light stimulation. (**c**) Light induced no response in wild-type mice. The number on top of the response map shows the fraction of laser intensity (3.6 mW). The red arrow indicates the fiber tip in the vascular pattern. The scale bar is 1 mm. The scale bar is also applied in (**d**–**f**). (**d**–**f**) Light-induced response pattern in VGAT-ChR2 mice (**d**), PV-ChR2 mice (**e**) and SST-ChR2 mice (**f**) at variable laser intensities. The number on top of the response map shows the fraction of laser intensity (3.6 mW). The dashed line indicates the lambdoid suture in the vascular pattern. The red rectangle shows the ROI near the fiber tip, and the black rectangle shows the ROI far from the fiber tip. (**g**) The spatial dynamics of the intrinsic signal induced by different laser intensities in VGAT-ChR2 mice. The ROI is outlined by a dashed green line on the vascular pattern. (**h**) The temporal dynamics of far (**left**) and near (**right**) responses. The blue line indicates the duration of the laser. The ROIs are outlined by black or red lines on the vascular pattern. (**i**,**j**) Summary of the responses of the three types of mice at 0.4 mW (near, F(2,11) = 7.231, *p* = 0.0099; far, F(2,11) = 0.7875, *p* = 0.479, one-way ANOVA) and 3.6 mW (near, F(2,11) = 8.12, *p* = 0.0068; far, F(2,11) = 9.238, *p* = 0.0044, one-way ANOVA). The bars represent the average. * *p* < 0.05, ** *p* < 0.01.

**Figure 2 ijms-24-05750-f002:**
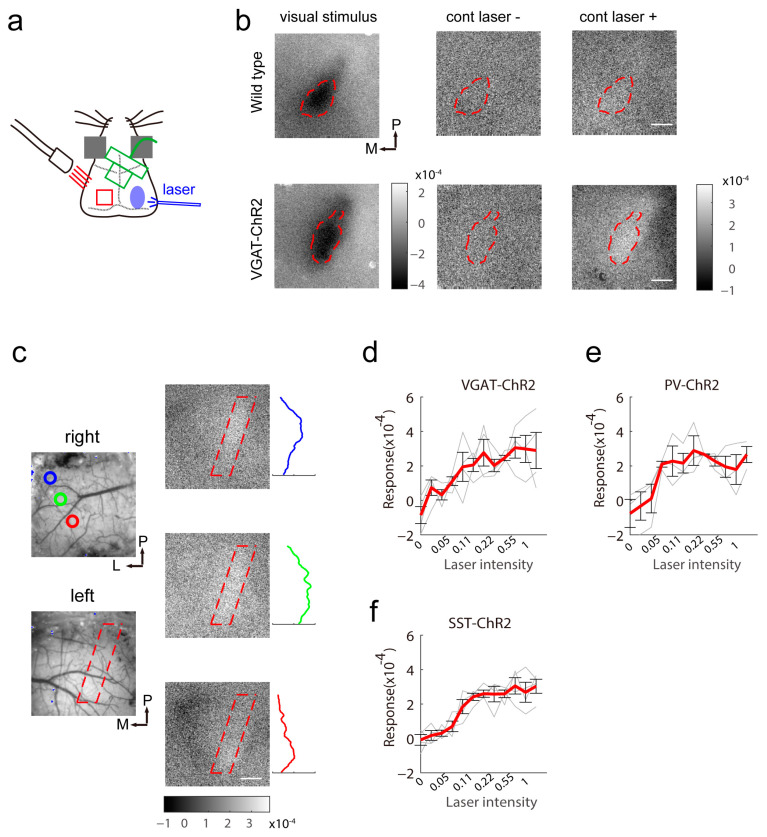
Reduced spontaneous activity elicited by contralateral light stimulation. (**a**) Schematic of intrinsic signal optical imaging and laser stimuli. The intrinsic signal optical imaging recorded the signal of the left visual cortex and light stimulation was located in the right visual cortex. (**b**) Response map induced by visual stimuli and contralateral light illumination in wild-type (**top**) and VGAT-ChR2 (**bottom**) mice (16 repeats). The binocular area was determined by the response evoked by a visual stimulus at the binocular receptive field (**left**). The spontaneous activity map with (**right**) or without (**middle**) contralateral light stimulation is illustrated. The scale bar is 0.2 mm. (**c**) The spontaneous response depression induced by contralateral light was topologically arranged in VGAT-ChR2 mice. The colored circles label the site for stimulation within the binocular area. The response map on the right shows the spatial activation pattern. The colored response curve was calculated from the region of interest outlined by the red dashed line in the response map. The scale bar is 0.2 mm. (**d**–**f**) The amplitude of the response of the left visual cortex depends on the laser intensity in VGAT-ChR2 mice (d, n = 4), PV-ChR2 mice (e, n = 3) and SST-ChR2 mice (f, n = 4).

**Figure 3 ijms-24-05750-f003:**
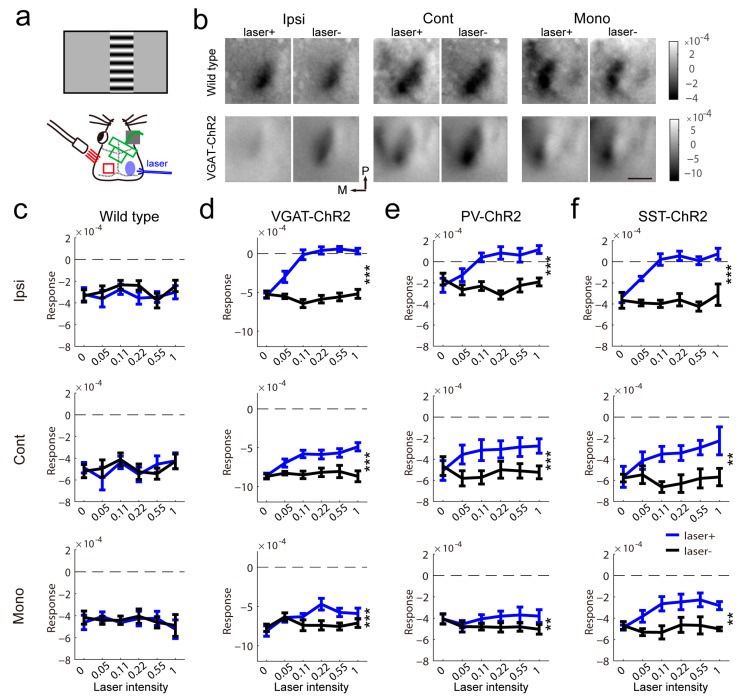
Modulation of visual response by contralateral inhibitory activation. (**a**) Schematic of intrinsic signal optical imaging, laser stimulation and visual stimuli. (**b**) The representative effect of laser illumination in the contralateral visual cortex on visually evoked responses in wild-type and VGAT-ChR2 mice. The mean of the response map induced by variable intensity is shown (96 repeats). The scale bar is 1 mm. (**c**–**f**) Summary of the effect in wild-type (**c**), VGAT-ChR2 (**d**), PV-ChR2 (**e**) and SST-ChR2 (**f**) mice (c, wild-type, n = 5; d, VGAT-ChR2, n = 8, Ipsi, F(1,7) = 71.03, *p* < 0.0001, Cont, F(1,7) = 41.89, *p* = 0.0003, Mono, F(1,7) = 109.8, *p* < 0.0001; e, PV-ChR2, n = 9, Ipsi, F(1,8) = 29.31, *p* = 0.0006, Cont, F(1,8) = 64.49, *p* < 0.0001, Mono, F(1,8) = 19.47, *p* = 0.0022; f, SST-ChR2, n = 4, Ipsi, F(1,3) = 451, *p* = 0.0002, Cont, F(1,3) = 95.32, *p* = 0.0023, Mono, F(1,3) = 42.01, *p* = 0.0075, two-way repeated measure ANOVA). ** *p* < 0.01, *** *p* < 0.001.

**Figure 4 ijms-24-05750-f004:**
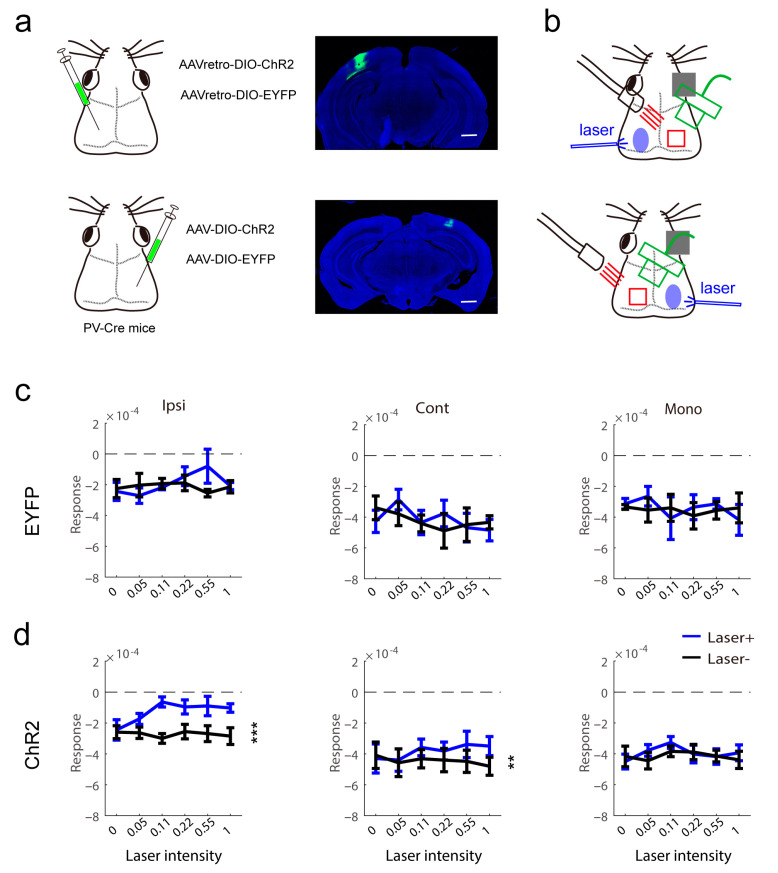
Modulation of visual response in virus-injected PV-Cre mice. (**a**) Virus injection and channelrhodopsin-2 protein expression in PV interneurons in the visual cortex. The scale bars are 2 mm (right). (**b**) Schematic of intrinsic signal optical imaging and laser stimuli. The intrinsic signal optical imaging recorded the signal of the left visual cortex and light stimulation was located in the right visual cortex. (**c**,**d**) The effect of light in contralateral light on measurement of visual response in control (c, n = 4) and PV (d, n = 9) mice (d, PV, Ipsi, F(1,8) = 33.18, *p* = 0.0004, Cont, F(1,8) = 18.13, *p* = 0.0028, Mono, F(1,8) = 3.528, *p* = 0.0972, two-way repeated measure ANOVA). ** *p* < 0.01, *** *p* < 0.001.

**Figure 5 ijms-24-05750-f005:**
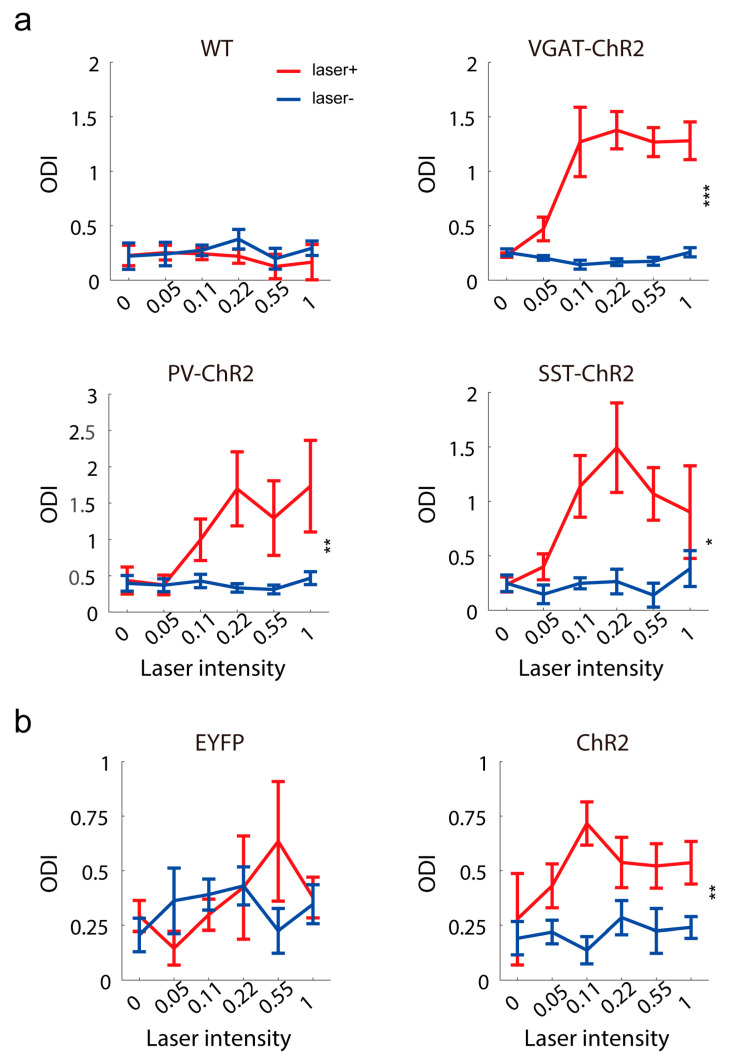
Ocular dominance changes induced by the activation of contralateral inhibitory neurons. (**a**) The ocular dominance index (ODI) with or without laser stimuli in wild-type (n = 5), VGAT-ChR2 (n = 8, F(1,7) = 36.63, *p* = 0.0004, two-way RM ANOVA), PV-ChR2 (n = 9, F(1,8) = 18.06, *p* = 0.0028, two-way RM ANOVA), SST-ChR2 (n = 4, F(1,3) = 20.02, *p* = 0.0208, two-way RM ANOVA) mice. (**b**) The ocular dominance index with or without laser stimuli in PV-Cre mice injected with virus expressing EYFP (n = 4) and ChR2 (n = 9) (F(1,8) = 14.49, *p* = 0.0052, two-way RM ANOVA). * *p* < 0.05, ** *p* < 0.01, *** *p* < 0.001.

**Figure 6 ijms-24-05750-f006:**
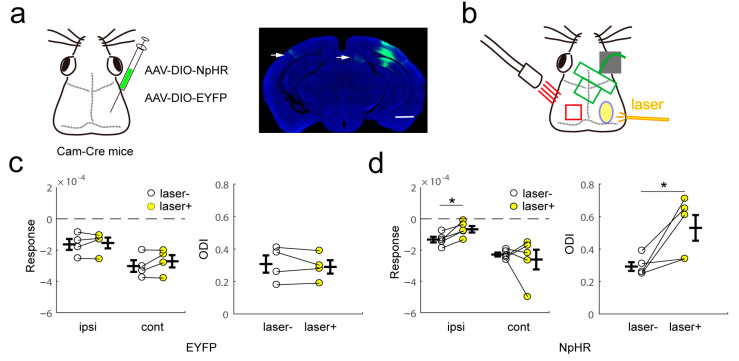
Ipsilateral eye response depression and ocular dominance change induced by optogenetic inhibition of contralateral excitatory neurons. (**a**) Virus injection (left) and halorhodopsin protein expression in excitatory neurons in the visual cortex (right). The scale bar is 2 mm (**right**). The white arrow shows the fiber right visual cortex. (**b**) Schematic of intrinsic signal optical imaging and laser stimuli. The intrinsic signal optical imaging recorded the signal of the left visual cortex, and light stimulation was located in the right visual cortex. (**c**,**d**) Response magnitude and ocular dominance index with or without laser stimuli in EYFP (**c**) and NpHR (**d**) mice (for d, ipsi, *p* = 0.0131, ODI, *p* = 0.0398, paired *t*-test). * *p* < 0.05.

## Data Availability

The data analyzed and presented in this study are available from the corresponding author on request.

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
