# Peer review of "Modulation of Visual Responses and Ocular Dominance by Contralateral Inhibitory Activation in the Mouse Visual Cortex"

_ijms, 2023, doi:10.3390/ijms24065750_

Round 1

Reviewer 1 Report

This manuscript covers an interesting topic. The authors have done a good job, but in some sections/points they need to carefully improve the manuscript. English needs another review, before final decision.

- The first sentence of the Introduction section is very long and unclear; please improve it.

- The aim of the present study is not described and the last section of the introduction is not clear.

- Lines 57 to 64 are not suitable in this section, because they report the methodology and the results.

- Delete “2.1. Subsection” and correct the titles of the subsection Results with their respective numbers; the same for the section of Materials and Methods.

- Please mention full names first and then their acronyms, not like in the current version.

- Put the respective figure when it is mentioned in the text (Results section) and not all at the end (therefore delete "2.2. Figures _ line 175")

- Create a new subsection for "Conclusions", mentioning the final conclusions of this study.

- For the titles of Materials and Methods, reformulate the titles, because a single word does not make sense for the respective paragraph when reading.

Author Response

This manuscript covers an interesting topic. The authors have done a good job, but in some sections/points they need to carefully improve the manuscript. English needs another review, before final decision.

- The first sentence of the Introduction section is very long and unclear; please improve it.

- The aim of the present study is not described and the last section of the introduction is not clear.

- Lines 57 to 64 are not suitable in this section, because they report the methodology and the results.

- Delete “2.1. Subsection” and correct the titles of the subsection Results with their respective numbers; the same for the section of Materials and Methods.

- Please mention full names first and then their acronyms, not like in the current version.

- Put the respective figure when it is mentioned in the text (Results section) and not all at the end (therefore delete "2.2. Figures _ line 175")

- Create a new subsection for "Conclusions", mentioning the final conclusions of this study.

- For the titles of Materials and Methods, reformulate the titles, because a single word does not make sense for the respective paragraph when reading.

We thank the reviewer for thinking this manuscript is interesting and the positive comment. We have revised the manuscript following the reviewer’ suggestions.

- The first sentence of the Introduction section is very long and unclear; please improve it.

We have re-written the first sentence as the following: “Long-range inputs are thought to modulate neuronal responses by supplying con-textual information. Running can double the visual response of visual cortex neurons by basal forebrain cholinergic innervation, while serotonergic input from the dorsal raphe nucleus suppresses spontaneous and evoked activity in the visual cortex [1, 2].”

- The aim of the present study is not described and the last section of the introduction is not clear.

To clarify the aim of this study and make it clear for the last section of the introduction, we have re-written this section as the following: ”To evaluate the effect of activation of different inhibitory neurons in one hemisphere on the visual response and ocular dominance of another hemisphere, we optogenetically activated distinctive interneurons in one hemisphere and recorded the response of the visual cortex in another hemisphere using intrinsic signal optical imaging. We found that optogenetic manipulation reduced the visually evoked response and shifted ocular dominance to the contralateral eye. Our findings revealed a transcallosal effect of activation of inhibitory neurons.”

- Lines 57 to 64 are not suitable in this section, because they report the methodology and the results.

According to the reviewer’s advice, we have shortened the description of the methodology and the results in this paragraph.

- Delete “2.1. Subsection” and correct the titles of the subsection Results with their respective numbers; the same for the section of Materials and Methods.

We have revised the titles of the subsections of the Results, also the Materials and Methods following the reviewer’s suggestions.

- Please mention full names first and then their acronyms, not like in the current version.

We have checked all acronyms in this manuscript and mentioned their full names first, such as GABA, VGAT, ChR2, PV, SST, AAV, ODI, and CaMKIIα.

- Put the respective figure when it is mentioned in the text (Results section) and not all at the end (therefore delete "2.2. Figures _ line 175")

We have added the respective figures in the results section and deleted “2.2. Figures _ line 175”.

- Create a new subsection for "Conclusions", mentioning the final conclusions of this study.

We have added the “Conclusions” subsection as following:” In this study, the effects on visual responses and ocular dominance of activation of contralateral inhibitory interneurons in the visual cortex were evaluated by intrinsic signal optical imaging. Optogenetic activation shows that different inhibitory neurons have distinctive response patterns. They all show inhibition of spontaneous activity in the contralateral visual cortex, and inhibition depends on the density of stimulations. Visually evoked responses were depressed by the activation of contralateral inhibitory neurons, and activation had the largest impact on the response from ipsilateral eyes. Thus, it shifts the ocular dominance to the contralateral eyes. Inhibition of excitatory neurons in the contralateral visual cortex reduced the ipsilateral eye response and moderately shifted the ocular dominance to the contralateral eye, suggesting that inhibition of excitatory neurons only partially mediates the effect of activation of inhibitory neurons. We conclude that activation of inhibitory neurons can modulate the visual response and ocular dominance in the contralateral visual cortex.”

- For the titles of Materials and Methods, reformulate the titles, because a single word does not make sense for the respective paragraph when reading.

We have reformulated the titles for the respective paragraph.

Reviewer 2 Report

Dear Editor,
I really appreciate the opportunity to review the manuscript ijms-2257630 entitled:
"Modulation of visual responses and ocular dominance by contralateral inhibitory activation in mouse visual cortex"

I commend the authors for describing this critical and timely issue. The paper is interesting and well-written; however, I would like to highlight some issues that merit revision:

I commend the authors for describing this critical and timely issue. The paper is interesting and well-written, and as a reviewer, I have no issues to highlight. Very good.

Author Response

Dear Editor,
I really appreciate the opportunity to review the manuscript ijms-2257630 entitled:
"Modulation of visual responses and ocular dominance by contralateral inhibitory activation in mouse visual cortex"

I commend the authors for describing this critical and timely issue. The paper is interesting and well-written; however, I would like to highlight some issues that merit revision:

I commend the authors for describing this critical and timely issue. The paper is interesting and well-written, and as a reviewer, I have no issues to highlight. Very good.

We thank the reviewer’s positive comments.

Reviewer 3 Report

The article wrote by Wu W et al., focuses on studying the effect of activation of inhibitory neurons in one hemisphere on the visual response and ocular dominance of another hemisphere, through the intrinsic signal optical imaging and optogenetics. The study pinpoints different response patterns are induced by optogenetic activation of different inhibitory neurons, not only affecting the cortex locally but also innervated the contralateral side transcallosally. The paper is well-written, the Authors’ addressed a very interesting topic using a specific animal model and optogenetic tools. The figures and the graphs have good resolution and very informative about the results showed, even if they should be better placed immediately after their description in the text, no needing a separated paragraph for figures by they need to be included in the Results section. Thus, the manuscript can be accepted in its present form after having addressed the minor revision listed here below.

Please check carefully how to insert Authors’ names and affiliations. Referred to IJMS Authors’ guidelines.

Abstract. The Authors are suggested to improve the Abstract, especially the first sentence, and the reason they used this methodology to investigate.

Line 15: there is a missing “s” in “stimulation”. Please check and revise.

Line 38: what does GABA stand for? Please specify the full name before giving the acronym

Line 40: an “s” is missing in “projection”. Please check and revise.

In general, there are some missing s in the manuscript. Please check and revise.

Line 66: please remove “2.1. Subsection”

Author Response

The article wrote by Wu W et al., focuses on studying the effect of activation of inhibitory neurons in one hemisphere on the visual response and ocular dominance of another hemisphere, through the intrinsic signal optical imaging and optogenetics. The study pinpoints different response patterns are induced by optogenetic activation of different inhibitory neurons, not only affecting the cortex locally but also innervated the contralateral side transcallosally. The paper is well-written, the Authors’ addressed a very interesting topic using a specific animal model and optogenetic tools. The figures and the graphs have good resolution and very informative about the results showed, even if they should be better placed immediately after their description in the text, no needing a separated paragraph for figures by they need to be included in the Results section. Thus, the manuscript can be accepted in its present form after having addressed the minor revision listed here below.

Please check carefully how to insert Authors’ names and affiliations. Referred to IJMS Authors’ guidelines.

Abstract. The Authors are suggested to improve the Abstract, especially the first sentence, and the reason they used this methodology to investigate.

Line 15: there is a missing “s” in “stimulation”. Please check and revise.

Line 38: what does GABA stand for? Please specify the full name before giving the acronym

Line 40: an “s” is missing in “projection”. Please check and revise.

In general, there are some missing s in the manuscript. Please check and revise.

Line 66: please remove “2.1. Subsection”

We thank the reviewer for the positive comments. We have revised the manuscript following the reviewer’ suggestions.

Please check carefully how to insert Authors’ names and affiliations. Referred to IJMS Authors’ guidelines.

We have revised the insertion according to IJMS Authors’ guidelines.

Abstract. The Authors are suggested to improve the Abstract, especially the first sentence, and the reason they used this methodology to investigate.

We have improved the abstract as following: “Both hemispheres connect with each other by excitatory callosal projections, and whether inhibitory interneurons, usually believed to have local innervation, engage in transcallosal activity modulation is unknown. Here, we used optogenetics in combination with cell type-specific channelrhodopsin-2 expression to activate different inhibitory neuron subpopulations in the visual cortex and recorded the response of the entire visual cortex using intrinsic signal optical imaging. We found that optogenetic stimulation of inhibitory neurons reduced spontaneous activity (increase in the reflection of illumination) in the binocular area of the contralateral hemi-sphere, although these stimulations had different local effects ipsilaterally. The activation of contralateral interneurons differentially affected both eye responses to visual stimuli and thus changed ocular dominance. Optogenetic silencing of excitatory neurons affects the ipsilateral eye response and ocular dominance in the contralateral cortex to a lesser extent. Our results revealed a transcallosal effect of interneuron activation in the mouse visual cortex.”

Line 15: there is a missing “s” in “stimulation”. Please check and revise.

We have checked and revised the missing “s”.

Line 38: what does GABA stand for? Please specify the full name before giving the acronym

GABA stands for gamma-aminobutyric acid. We have checked and given the full name of the acronyms in the manuscript.

Line 40: an “s” is missing in “projection”. Please check and revise.

In general, there are some missing s in the manuscript. Please check and revise.

 We have checked and revised the missing “s”.

Line 66: please remove “2.1. Subsection”

We have removed “2.1. Subsection”.

Round 2

Reviewer 1 Report

The authors have improved and changed the points respecting the suggestions! The revised version is now suitable for publication!